# Endolysosomal Ca^2+^ Signalling and Cancer Hallmarks: Two-Pore Channels on the Move, TRPML1 Lags Behind!

**DOI:** 10.3390/cancers11010027

**Published:** 2018-12-27

**Authors:** Pawan Faris, Mudhir Shekha, Daniela Montagna, Germano Guerra, Francesco Moccia

**Affiliations:** 1Laboratory of General Physiology, Department of Biology and Biotechnology “L. Spallanzani”, University of Pavia, 27100 Pavia, Italy; pawan.faris@yahoo.com; 2Research Centre, Salahaddin University-Erbil, Erbil 44001, Kurdistan-Region of Iraq, Iraq; mudhir.shekha@su.edu.krd; 3Department of Pathological Analysis, College of Science, Knowledge University, Erbil 074016, Kurdistan-Region of Iraq, Iraq; 4Laboratory of Immunology Transplantation, Foundation IRCCS Policlinico San Matteo, 27100 Pavia, Italy; d.montagna@smatteo.pv.it; 5Department of Sciences Clinic-Surgical, Diagnostic and Pediatric, University of Pavia, 27100 Pavia, Italy; 6Department of Medicine and Health Sciences “Vincenzo Tiberio”, University of Molise, 86100 Campobasso, Italy; germano.guerra@unimol.it

**Keywords:** endolysosomal Ca^2+^ signaling, cancer, TRPML1, TPCs, NAADP, proliferation, metastasis, angiogenesis

## Abstract

The acidic vesicles of the endolysosomal (EL) system are emerging as an intracellular Ca^2+^ store implicated in the regulation of multiple cellular functions. The EL Ca^2+^ store releases Ca^2+^ through a variety of Ca^2+^-permeable channels, including Transient Receptor Potential (TRP) Mucolipin 1-3 (TRPML1-3) and two-pore channels 1-2 (TPC1-2), whereas EL Ca^2+^ refilling is sustained by the proton gradient across the EL membrane and/or by the endoplasmic reticulum (ER). EL Ca^2+^ signals may be either spatially restricted to control vesicle trafficking, autophagy and membrane repair or may be amplified into a global Ca^2+^ signal through the Ca^2+^-dependent recruitment of ER-embedded channels. Emerging evidence suggested that nicotinic acid adenine dinucleotide phosphate (NAADP)-gated TPCs sustain multiple cancer hallmarks, such as migration, invasiveness and angiogenesis. Herein, we first survey the EL Ca^2+^ refilling and release mechanisms and then focus on the oncogenic role of EL Ca^2+^ signaling. While the evidence in favor of TRPML1 involvement in neoplastic transformation is yet to be clearly provided, TPCs are emerging as an alternative target for anticancer therapies.

## 1. Introduction

Intracellular Ca^2+^ signals regulate a plethora of cellular functions, including exocytosis, proliferation, migration, differentiation, gene transcription, bioenergetics and survival [1,2]. The resting intracellular Ca^2+^ concentration ([Ca^2+^]_i_) is maintained at ≈100 nM by the activity of a number of Ca^2+^ transporting systems which extrude Ca^2+^ across the plasma membrane (PM), such as the Plasma Membrane Ca^2+^-ATPase (PMCA) and the Na^+^/Ca^2+^ exchanger (NCX), or sequester Ca^2+^ within the lumen of endogenous organelles, such as the Sarco-Endoplasmic Reticulum Ca^2+^-ATPase (SERCA), the mitochondrial Ca^2+^ uniporter (MCU) and the Secretory Pathway Ca^2+^-ATPase (SPCA), which sequesters Ca^2+^ into the Golgi lumen [1,2,3,4]. An increase in intracellular Ca^2+^ concentration ([Ca^2+^]_i_) is produced by the opening of Ca^2+^-permeable channels, which are located either on the PM or on the membrane of endogenous organelles, in response to extracellular stimulation. Typically, extracellular agonists, including growth factors, cytokines, neurotransmitters and hormones, bind to their specific tyrosine kinase receptors (TKRs) or G_q_-protein coupled receptors (GPCRs), thereby recruiting, respectively, phospholipase Cγ (PLCγ) and PLCβ, which in turn cleave phosphatidylinositol-4,5-bisphosphate (PIP_2_) into inositol-1,4,5-trisphosphate (InsP_3_) and diacyl-glycerol (DAG) [1]. Subsequently, InsP_3_ primes immobile InsP_3_ receptors (InsP_3_Rs), which reside at juxtaposed endoplasmic reticulum (ER)-PM junctions, to respond to cytosolic Ca^2+^ according to the process of Ca^2+^-induced Ca^2+^ release (CICR), which enables to spatially restricted Ca^2+^ signals to spread across the cytoplasm as a regenerative Ca^2+^ wave [5,6]. Moreover, InsP_3_-induced Ca^2+^ release can be further amplified by the recruitment of adjoining ryanodine receptors (RyRs), which are also sensitive to CICR [7]. Additionally, InsP_3_-induced Ca^2+^ release causes a large drop in the ER Ca^2+^ concentration ([Ca^2+^]_ER_) which triggers a signaling cascade leading to the opening of a Ca^2+^-permeable pathway on the PM, known as store-operated Ca^2+^ entry (SOCE) [8,9]. SOCE is activated when ER Ca^2+^ dissociates from the luminal EF hands domain of STIM1 (Stromal Interaction Molecule 1), the sensor of [Ca^2+^]_ER_, which undergoes a dramatic conformational change and sub-cellular repositioning. The depletion of ER Ca^2+^ induces STIM1 to oligomerize and translocate from its reticular location in the bulk ER to specialized ER-PM junctions (≈20 nm) where it forms STIM1 *puncta* and interacts with the pore-forming subunit of store-operated channels (SOCs), Orai1 [8]. The dynamic interplay between InsP_3_-induced ER Ca^2+^ release and SOCE gives raise to distinct spatio-temporally complex Ca^2+^ signatures, ranging from local Ca^2+^ puffs to repetitive global Ca^2+^ oscillations, that selectively regulate different cellular functions [2,10,11].

The ER is by far the largest and best studied intracellular Ca^2+^ reservoir engaged by extracellular stimuli, possessing a complete repertoire of Ca^2+^ pumps, i.e., SERCA, to effect filling, luminal Ca^2+^-binding proteins, e.g., calnexin and calreticulin, to effect buffering and Ca^2+^-permeable channels, i.e., InsP_3_Rs and RyRs, to effect release [7,12]. Nevertheless, the ER is not the sole endogenous Ca^2+^ store in mammalian cells. It has been shown that also the acidic vesicles of the endo-lysosomal (EL) system serve as an additional Ca^2+^ reservoir that is recruited by extracellular ligands through the generation of the novel Ca^2+^-releasing second messenger, nicotinic acid adenine dinucleotide phosphate (NAADP) [12,13,14,15]. The battery of Ca^2+^ refilling and release mechanisms endowed to acidic EL vesicles is emerging as a crucial signalling network to finely tune local processes, such as membrane fusion and individualized vesicular movements [16]. In addition, the establishment of EL-ER membrane contact sites (MCSs) enables EL Ca^2+^ release to trigger cytosolic Ca^2+^ waves by recruiting ER-resident InsP_3_Rs and RyRs through the CICR process [17,18].

Herein, we discuss the emerging role of EL Ca^2+^ signalling in tumorigenesis [19,20,21]. We first illustrate the basic mechanisms of EL Ca^2+^ refilling and release by focusing on the wealth of Ca^2+^-permeable channels that effect EL Ca^2+^ mobilization, i.e., Transient Receptor Potential (TRP) Mucolipin 1-3 (TRPML1-3), two-pore channels 1-2 (TPC1-2), TRP Melastatin 2 (TRPM2), TRP Ankyrin 1 (TRPA1), P2X4 receptors and voltage-gated Ca^2+^ channels (Figure 1). Then, we describe how TPCs control several cancer hallmarks, including migration, invasiveness and angiogenesis, and the potential tumorigenic function of TRPML1. Finally, we speculate about the possibility to interfere with the EL Ca^2+^ signalling machinery to design alternative anticancer treatments, by focusing on TPCs as emerging molecular targets.

## 2. The Endolysosomal Ca^2+^ Store: Functions and Mechanisms of Ca^2+^ Refilling and Release

### 2.1. The Function of Endosomes and Lysosomes: Their Emerging Role in Ca^2+^ Signaling

The EL Ca^2+^ store belongs to the emerging family of acidic Ca^2+^ stores (Figure 2), which is widespread across the phylogenetic tree and includes acidocalcisomes, vacuoles, lysosome-related organelles, secretory granules [12,15], and the Golgi complex, that has been recently reviewed [22,23]. Endocytosis is the mechanism whereby extracellular solutes, macromolecules, and PM components are internalized and redirected towards the lysosomes through PM invagination followed by the formation of vesicles and vacuoles through membrane fission [24] (Figure 2). Once formed at the PM, early endosomes (EEs) are weakly acidic (pH 6.8–5.9) and, although they were supposed to be on par with the extracellular Ca^2+^ concentration [12], they contain a relatively low amount of Ca^2+^ (≈2–3 μM) [25,26] (Figure 2). Nevertheless, their intraluminal Ca^2+^ concentration is still at least an order of magnitude higher than that of resting [Ca^2+^]_i_ [12] and, therefore, EEs may serve as endogenous Ca^2+^ reservoir. While the majority of EEs is recycled back to the PM within about 10–15 min [24], a small fraction of EEs progressively acidify (pH 4.8–6.0) due to the activity of a vacuolar type H^+^-ATPase (v-ATPase) and gradually mature into late endosomes (LEs) (Figure 2), also known as multivesicular bodies (MVBs), which undergo a remarkable increase in their intraluminal Ca^2+^ concentration before fusing with lysosomes [24,26,27]. Endosomes occupy only a small percentage of the total cell volume, i.e., (0.65–2%) [24], although they are highly mobile (0.2–0.5 μm/sec) [28]: therefore, they are nicely suited to trigger local increases in [Ca^2+^]_i_ [12,26], although their intimate association with ER cisternae could result in global intracellular Ca^2+^ waves (see the Section 2.4.2.1).

Lysosomes are dynamic membrane-bound organelles that constitute up to 5% of the intracellular volume and show remarkable heterogeneity in size (100–500 nm in diameter), morphology and distribution [29]. Lysosomes were long regarded as mere waste bags by serving as the terminal degradative compartment recycling extracellular particles, cellular macromolecules and damaged organelles delivered via endocytosis, phagocytosis and autophagy [29,30] (Figure 2). Lysosomal function requires the establishment of a highly acidic pH (4.5–5.0), which is generated by an increase in v-ATPase activity on the lysosomal membrane and is supported by a resting lysosomal membrane potential (ΔΨ) ranging between −20 mV and −40 mV (lumen positive) [29,31]. This acidic pH is indispensable to support the activity of more than 60 hydrolases, such as nucleases, glycosidases, phosphatases, sulfatases, lipases and proteases, which are primarily involved in the digestion of proteins, polysaccharides, and complex lipids, into their building-block constituents: respectively, amino acids (AAs), monosaccharides, and free fatty acids [29,30]. Nevertheless, the last ten years witnessed a tremendous increase in our knowledge about the effective role of lysosomes in driving cellular fate and behavior. It is now acknowledged that lysosomes also contain 25 transmembrane proteins that enable these organelles to serve as a sophisticated regulatory hub which integrates multiple signals to finely tune cell metabolism, division and growth [32,33]. The lysosomal membrane plays a crucial role in nutrient sensing and metabolic regulation by hosting the mechanistic target of rapamycin complex 1 (mTORC1) and the energy-sensing kinase, AMP-activated protein kinase (AMPK) [32,34], as well as a number of proteins involved in mTORC1 regulation, including the Rag GTPases, and a pentameric complex termed Ragulator or late endosomal/lysosomal adaptor, mitogen-activated protein kinases (MAPK) and mTOR activator 1 (LAMTOR) [34]. In addition, lysosomes may establish a bidirectional Ca^2+^ exchange with the ER, thereby regulating pleoiotropic functions across the phylogenetic tree [18,35,36].

Accordingly, lysosomal lumen contains a high amount of Ca^2+^, ≈600 μM [37,38], that is within the same range as that reported in the ER, is mobilized by the second messenger, NAADP [13], and may sustain both local and global elevations in [Ca^2+^]_i_ [12,13]. The presence of releasable Ca^2+^ within the lysosomes has been demonstrated by treating cells with the lysosomotropic agent, Gly-Phe-naphthylamide (GPN), a substrate of cathepsin C which is selectively expressed in lysosomes, but not in prelysosomal endocytic vacuoles [12]. GPN is degraded by cathepsin C, thereby causing osmotic swelling of the lysosomes and inducing a discernible increase in [Ca^2+^]_i_ in a wide number of cells types [12,39,40,41]. Local lysosomal Ca^2+^ signals regulate normal trafficking, recycling, and vesicular fusion events with autophagosomes and LEs [16,29]. It has been suggested that the ubiquitously expressed C2-domain containing synaptotagmin isoform VII (Syt VII) and the EF-hands domain containing calmodulin (CaM) and apoptosis-linked gene-2 (ALG-2) function as the Ca^2+^-sensors that, respectively, mediate lysosomal exocytosis and LE-lysosomes fusion [29,42]. In addition, spatially-restricted lysosomal Ca^2+^ release is required to promote the starvation-induced nuclear translocation of TFEB [43], as described in the next paragraph.

### 2.2. Lysosomal Ca^2+^ Signaling Controls Lysosome Biogenesis and Autophagy

The master nutrient-responsive kinase, mTORC1, controls the cellular response to nutrient deficiency/availability by perceiving signals arising from growth factors, nutrients, and energy [30,44,45]. In the presence of nutrients, such as amino acids, lysosomes are positioned beneath the PM and mTORC1 is easily recruited to the lysosomal surface by the physical interaction with some of their transmembrane proteins, including Rag GTPases and the Ragulator complex and the v-ATPase. Herein, mTORC1 is activated by the small GTPase Rheb and phosphorylates its downstream targets, thereby stimulating cell growth and metabolism through the activation of biosynthetic pathways (including protein, lipid and nucleotide synthesis) and the inhibition of catabolic processes [44,45,46]. In particular, mTORC1 phosphorylates S6K and 4E-BP1 to promote protein synthesis, while it phosphorylates ULK1 and the transcription factor EB (TFEB) to inhibit autophagosome formation and lysosomal biogenesis, respectively [33,45,46]. Accordingly, phosphorylated TFEB is retained in the cytoplasm and cannot activate the catabolic transcriptional program that drives autophagy [47]. In contrast, starvation causes perinuclear clustering of lysosomes, which is caused by an increase in intracellular pH, and inactivates mTORC1 through multiple mechanisms, including the depletion of nutrients and a phosphorylation cascade triggered by AMPK. mTORC1 inactivation, in turn, interrupts the biosynthetic pathways and leads to ULK1 and TFEB dephosphorylation in a lysosomal Ca^2+^-dependent manner [44,45,46] (Figure 3). As a consequence, TFEB is now free to translocate into the nucleus, where it physically binds to the CLEAR (coordinated lysosomal expression and regulation gene network) elements, thereby inducing the transcription of a gene network involved in lysosomal biogenesis and autophagy [47]. In particular, the CLEAR element is targeted by a family of basic helix-loop-helix transcription factors, known as the MiT/TFE proteins, which include TFEB, TFEC, TFE3, and MITF [47,48,49]. TFEB, as well as MITF and TFE3, directly bind to CLEAR elements, thereby inducing the expression of target genes [33]. For instance, TFEB elicits a remarkable expansion of the lysosomal compartment, in terms of size, number, and protein content of lysosomal vesicles [47]. Furthermore, TFEB stimulates the expression of multiple autophagic proteins, including autophagosomal initiation (NRBF2 and BECN1), elongation (ATG9b, GABARAP, WIPI2), substrate capture (SQSTM1), and autophagosomal trafficking and fusion with lysosomes (UVRAG) [48,50].

### 2.3. Ca^2+^ Uptake Mechanisms in the Lysosomes

As discussed in [14], endocytosis contributes to only (≈2–3 μM) of the luminal free Ca^2+^ concentration within the lysosomes [26,37]. It turns out that lysosomes require a Ca^2+^ transporter or exchanger to sequester cytosolic Ca^2+^ into their lumen just as SERCA fills the ER with Ca^2^+ [14,51]. It is largely accepted that lysosomal Ca^2+^ refilling is driven by the proton-motive force as manoeuvres that dissipate the H^+^ gradient deplete this Ca^2+^ store or prevent its refilling [14]. Lysosomal Ca^2+^ storage may be impaired by direct alkalization of the lumen with NH_4_Cl or with protonophores, such as monensin and nigericin, that abolish the H^+^ gradient [14,37]. Additionally, lysosomal Ca^2+^ uptake can be impaired by targeting the v-ATPase with selective inhibitors, such as bafilomycin A1 and concanamycin [14,40,52,53], provided that sufficient lysosomal Ca^2+^ leak is available [14,52]. A Ca^2+^/H^+^ exchanger (CAX) was first identified in vacuoles, which serve as acidic Ca^2+^ stores in yeast and plants [14], and, more recently, in nonplacental mammalian cells [54] (Figure 1). Nevertheless, the molecular identification of the CAX in placental mammals remains still elusive [16]. Alternately, lysosomal Ca^2+^ refilling could occur through a Na^+^/H^+^ exchange coupled in series with a Na^+^/Ca^2+^ exchanger (yet to be demonstrated) or through a P-type Ca^2+^-ATPase, which countertransports H^+^ and would, therefore, be sensitive to the acidic lysosomal lumen [14,55]. However, this latter mechanism has been unequivocally shown only in human platelets, where SERCA3 is primarily responsible for Ca^2+^ accumulation within acidic organelles [56].

Remarkably, a recent series of studies disclosed that lysosomes may sequester ER Ca^2+^ released through InsP_3_Rs in response to physiological stimuli [57,58]. Moreover, it has been suggested that the basal production of InsP_3_ could drive lysosomal Ca^2+^ refilling in an InsP_3_Rs-dependent, but pH-independent, manner in HEK293 cells [59]. It should, however, be pointed out that a previous investigation demonstrated that the H^+^ gradient was necessary for lysosomes to curb InsP_3_Rs-mediated Ca^2+^ signals in the same cell type [57,58]. As mentioned earlier, v-ATPase inhibitors may reliably deplete the lysosomal Ca^2+^ store only in the presence of an adequate Ca^2+^ leak. If Ca^2+^ uptake is effectively driven by a pH-dependent mechanism, but the resting lysosomal Ca^2+^ permeability is low, bafilomycin A1 and/or concanamycin will require longer time to cause a robust drop in the intraluminal free Ca^2+^ concentration, as elegantly discussed in [14]. Notably, HEK293 cells were preincubated in the presence of bafilomycin A1 for 1 h in [57,58] but only for less than 10 min in [59]. Of note, an independent study disclosed that lysosomal Ca^2+^ refilling requires an intact ER Ca^2+^ pool and is impaired by collapsing the H^+^ gradient in the widely employed cell line, HeLa cells [40]. An additional Ca^2+^ source for lysosomal refilling could be provided by SOCE, which replenishes their intraluminal Ca^2+^ content in peripheral lysosomes upon TFEB overexpression [60]. Collectively, these studies strongly hint at the H^+^ gradient as the main driver of lysosomal Ca^2+^ uptake, but the molecular identify of the transporter(s) responsible for lysosomal Ca^2+^ refilling in mammalian cells is still far from being elucidated. The role played by InsP_3_Rs in lysosomal Ca^2+^ refilling also deserves further investigations.

### 2.4. Endoysosomal Ca^2+^ Release Channels

The EL Ca^2+^ store express a variety of Ca^2+^-permeable channels that may result in both local and global intracellular Ca^2+^ signals. These include several members of the TRP superfamily of non-selective cation channels, such as TRPML1-3, TRPM2, and TRPA1, NAADP-gated TPC1-2, ATP-gated ionotropic P2X4 receptors and voltage-gated Ca^2+^ channels [12,14,15,16,19,21,42] (Figure 1). Moreover, the EL Ca^2+^ store contains a variety of Ca^2+^-impermeable channels that regulate Ca^2+^ homeostasis by modulating the H^+^ gradient and/or the lysosomal ΔΨ. These include ClC chloride channels (Cl^−^/H^+^ exchanger), big conductance Ca^2+^-activated K^+^ channels (BK, KCa1.1, MaxiK) and TMEM175 [42,61]. Notably, BK channels modulate ΔΨ, thereby controlling InsP_3_-dependent lysosomal Ca^2+^ refilling [62] and TRPML1-mediated Ca^2+^ release [63].

#### 2.4.1. TRPML1-3 Channels

The mucolipin subfamily of TRP proteins comprises three EL non-selective cation channels: TRPML1, TRPML2 and TRPML3 [64,65]. TRPML1, the founding member of this subfamily, was named after the channelopathy, mucolipidosis type IV (MLIV), caused by mutations in the TRPML1 gene (*MCOLN1*) [66]. Similar to all TRP channels [67], each TRPML is composed of 4 subunits which possess 6 transmembrane (TM) spanning domains with cytosolic NH_2_- and COOH-terminal tails; moreover, the structure of TRPML proteins presents a pore-forming re-entrant loop between TM5 and TM6, a large intraluminal loop between TM1 and TM2 and various cytosolic regulatory domains [64,65,66]. TRPML channels localize most exclusively to the EL compartment due to the presence of EL targeting sequences (D/EXXXLL/I) in their NH_2_- and COOH-terminal tails, although these dileucine motifs are absent in TRPML3 [15,66]. TRPML1 is widely distributed within the later vesicles of the endocytic pathway, such as LEs and lysosomes, TRPML2 resides at recycling endosomes, LE/lysosomes and PM, and TRPML3 mainly localizes to the EL system and PM [12,66] (Figure 2). However, it has been proposed that TRPML3 is retained within the acidic vesicles by the interaction with TRPML1 [66,68]. Moreover, TRPML1 may also reach the PM either upon delivery from the Golgi apparatus through the biosynthetic route or upon lysosomal exocytosis [64,69,70].

Excellent reviews have recently covered the molecular architecture, biophysical properties and role of TRPML channels [19,21,61,66]. Herein, we briefly recall that TRPML1 is permeable to Ca^2+^, Na^+^, Fe^2+^, Mg^2+^, and K^+^ and displays an inwardly-rectifying I/V relationship, which indicates that it facilitates the efflux of EL cations into the cytosol [21,71,72]. TRPML1 is not permeable to H^+^ [71], although the genetic loss of TRPML1 may cause a dramatic loss in intraluminal pH [73] and TRPML1 is finely tuned by lysosomal pH [21,74]. Elucidation of its crystal structure revealed that the large intraluminal loop between TM1 and TM2 contains 12 aspartate residues that can be protonated at low pH, thereby favoring channel activity [75]. Conversely, at neutral pH (7.4) these aspartate reside retain their negative charge and cause the so-called Ca^2+^ block [75], as also demonstrated in [76]. TRPML1 may be physiologically gated by phosphatidylinositol-3,5-bisphosphate (PI(3,5)P_2_), a phosphoinositide which is particularly abundant in LEs and lysosomes and is able to bind to the NH_2_-terminal tail of the channel [77,78]. Conversely, TRPML1 is inhibited by phosphatidylinositol-4,5-bisphosphate (PI(4,5)P_2_), which is likely to displace PI(3,5)P_2_ from its binding sites at the NH_2_-tail, but is more enriched at PM [78]. In addition, TRPML1 may be activated by reactive oxygen species (ROS), thereby serving as the lysosomal sensor of oxidative stress [79]. TRPML1-mediated Ca^2+^ release regulates a variety of Ca^2+^-dependent processes, including LE-lysosome fusion, lysosomal exocytosis, and membrane repair [16,29,80,81]. Furthermore, TRPML1-mediated lysosomal Ca^2+^ signals finely control autophagy by modulating mTORC1 activity TFEB localization, and autophagome-lysosome fusion [64]. It has been proposed that, when mTORC1 translocates onto the lysosomal surface in full-nutrient conditions, TRPML1-mediated local Ca^2+^ release supports Rheb-mediated mTORC1 activation to suppress autophagy in a calmodulin (CaM)-dependent manner [82]. As mentioned earlier, once activated, mTORC1 directly phosphorylates TFEB on two critical serine residues (Serine 142 and Serine 211), thereby causing its binding to the scaffold 14-3-3 proteins and its retention into the cytoplasm [33,83].

Conversely, in conditions of nutrient starvation, mTORC1 is inactivated and dissociates from the lysosomal surface [33,45,46]. Moreover, starvation causes a spatially-restricted increase in [Ca^2+^]_i_ concentration by further mobilizing lysosomal Ca^2+^ through TRPML1 [43]: these TRPML1-mediated local Ca^2+^ signals recruit calcineurin (CaN) to dephosphorylate TFEB, which dissociates from 14-3-3 proteins and shuttles into the nucleus to trigger the transcriptional program driving lysosomal biogenesis and autophagy [47,50,70] (Figure 3).

Notably, TRPML1 is also under the transcriptional control of TFEB, which results in the establishment of a feed-forward loop that ensures lysosomal adaptation to prolonged stress and lack of nutrients [84]. It should, however, be pointed out that starvation-induced TRPML1-mediated Ca^2+^ release could also restore mTORC1 activity during prolonged starvation as an adaptive mechanism to maintain protein synthesis [85]. While starvation causes a reduction in lysosomal PI(3,5)P_2_ levels [64], it stimulates ROS production [81], which could deliver the gating signal to TRPML1 and induce lysosomal-to-nucleus communication through CaN-dependent TFEB dephosphorylation [79,82]. Moreover, starvation induces the PI 5-phosphatase, OCRL, to translocate from endosomes to lysosomes and dephosphorylate PI(4,5)P_2_ to PI4P, thereby sustaining TRPML1 activation and the progression of the autophagic process [64,86,87]. Finally, it has recently been shown that PI(4,5)P_2_-induced TRPML1 activation following autophagy induction stimulates the Ca^2+^-dependent centripetal repositioning of lysosomes towards the perinuclear region, where autophagosomes accumulate [88]. TRPML1-induced Ca^2+^ release recruits the EF-hand-containing Ca^2+^ sensor, ALG-2, to physically associate with the minus-end-directed dynactin-dynein motor, which is the crucial step to ensure lysosomal relocalization [88].

Similar to TRPML1, TRPML2 is permeable to Na^+^, K^+^, Ca^2+^ and Fe^2+^, displays an inwardly-rectifying I/V relationship and is gated by H^+^ and PI(3,5)P_2_ [71,77,89]. It has been suggested that TRPML2 is recycled from PM to the endosome via clathrin-mediated endocytosis [68]. Thereafter, TRPML2 promotes the activation of the small GTPase, ADP-ribosylation factor 6 (Arf6), thereby regulating recycling of glycosylphosphatidylinositol-anchored proteins (GPI-APs), including CD59 [90]. In addition, TRPML2 may associate into functional heteromultimers either with TRPML1 or TRPML3, which present distinct biophysical properties compared to homomeric channels and could control lysosomal integrity and starvation-induced autophagy [91,92,93].

TRPML3 is a Ca^2+^ channel that is permeable also to monovalent cations (Na^+^ > K^+^ ≫ Cs^+^), but is inhibited by high cytosolic Na^+^, and shows an inwardly-rectifying I/V relationship [94,95,96,97]. Similar to TRPML1, TRPML3 is gated by PI(3,5)P_2_ while it is not clear whether it is inhibited by PI(4,5)P_2_ [98]. Nevertheless, TRPML3 is blocked, rather than potentiated, by lysosomal acidification due to the protonation of a string of three histidine residues (H252, H273, H283) within the large intraluminal loop between TM1 and TM2 [96]. As TRPML3 is blocked by Na^+^ and H^+^, which are more abundant in the lysosomal lumen [29], it has been suggested that it mediates Ca^2+^ release from recycling endosomes and EEs, but not from LEs/lysosomes [21,99]. However, TRPML3 could become activated by lysosomal damage, stress, or pathogen infection, which dissipate the H^+^ gradient and could lead to TRPML1 closure [21,61,100]. As mentioned earlier, TRPML3 is not targeted to the EL system by the classical dileucine motifs at the NH_2_- and COOH-terminal tails described above, but it could be redirected to EL vesicle by an EXXLL motif in the NH_2_ terminus [99]. A number of recent studies hinted at a role for TRPML3 in endocytosis, membrane trafficking, and autophagy [80]. For instance, TRPML3 drives both constitutive (i.e., transferrin) and regulated (i.e., epidermal growth factor [EGF] and EGF receptor [EGFR]) endocytosis most likely though a local increase in [Ca^2+^]_i_ [101]. Moreover, starvation and cellular stress favor TRPML3 recruitment to autophagosomes, thereby exacerbating the autophagic response [101]. Subsequent work revealed that TRPML3 interacts with GATE16, a mammalian ATG8 homologue, at the autophagosome and extra-autophagosomal organellar membranes to regulate the later phases of autophagosome biogenesis, which include fusion [102]. In this view, a recent investigation suggested that distinct TRPML channels could play different roles in lysosomal biogenesis showing that TRPML1-dependent Ca^2+^ release stimulates lysosomal fission and reformation in a CaM-dependent manner [80,102].

#### 2.4.2. TPC1-2 Channels: Structure, Gating Mechanisms and Controversies

Two-pore channels (TPCs) belong to the superfamily of voltage-gated ion channels [103]. Voltage-gated ion channels present a modular structure consisting of combinations of pore-forming regions with two TM α-helices and voltage sensors with four TM α-helices, which may be concatenated to form six TM domains [104]. TPCs were originally identified in rats [105] and *Arabidopsis thaliana* [106] based with their high sequence homology with the α subunit of voltage-gated Na^+^ (Na_V_) and Ca^2+^ (Ca_V_) channels. TPCs are so-named because each subunit is modularly organized into two domains, each containing 6 TM α-elices with the putative pore loops between TM5-TM6 and TM11-TM12, a large cytosolic linker between the two domains and cytosolic NH_2_- and COOH-termini [107,108,109]. Two TPC subunits dimerize into homo- or heterodimers to form a functional Ca^2+^-releasing channel. Therefore, TPCs represent the molecular intermediate in the evolutionary transition from tetrameric 1-domain (6 TM) channels, such as the α subunit of voltage-gated K^+^ channels (K_V_) or TRP channels, to monomeric 4-domain (4 × 6TM) channels, such as Na_V_ and Ca_V_ [103,110,111].

Three isoforms exist of this family: TPC1, TPC2 and TPC3, which are encoded by *TPCN1*, *TPCN2* and *TPCN3*, respectively. However, TPC3 is not present in mice, rats and primates, although it is expressed in other mammals, such as cats, dogs and chickens [103,112,113]. Two groups recently reported about the crystal structure of TPC1 from *Arabidopsis thaliana* (AtTPC1), which confirmed the membrane topology predicted on the basis of its amino acid sequence [114,115]. For instance, the X-ray structure of AtTPC1 obtained at 2.87 Å resolution revealed that each subunit comprises two asymmetrical Shaker-like 6-TM domains connected by a large cytosolic EF-hands domain. These subunits assemble into a central quasi-tetrameric channel with each polypeptide displaying two voltage-sensing domains, two-pore domains and activation gates [114]. Subsequently, the electron cryo-microscopy structure of mouse TPC1 (3.4 Å resolution) did not reveal any major difference with the atomic structure of AtTPC1, although Ca^2+^ cannot bind the EF-hands domain located within the cytosolic linker as this lacks the acidic residues responsible for this interaction. Moreover, the COOH-terminal tail is longer and adopts a horseshoe-shaped arrangement with four α-helices and two β-strands and physically associates with the EF-hands domain, thereby providing a remarkable steric hindrance [116].

TPC1 is widely distributed across the EL system, i.e., recycling endosomes, EEs, LEs and lysosomes, while TPC2 mainly resides at LEs and lysosomes [117,118,119] (Figure 2). The EL distribution of TPC1-2 depends on a dileucin motif at their NH_2_-terminal tails [120]. Accordingly, it has been demonstrated that TPCs finely tune EL membrane traffic and autophagy through local Ca^2+^ signals [80,103]. Furthermore, TPCs may trigger global, often repetitive, elevations in [Ca^2+^]_i_ that drive a variety of cellular processes, ranging from fertilization to muscle contraction [13,52,121,122]. The gating mechanisms of TPCs and their permeability properties have engendered a durable controversy that is yet to be fully resolved [103,123,124,125,126]. TPC1-2 were first identified as the long-sought target for the intracellular second messenger, NAADP, which is synthesized in response to multiple extracellular stimuli by the ADP-ribosyl cyclase, CD38 [127]. A wealth of robust evidences clearly converged on this evidence. First, overexpression of TPCs potentiated NAADP-induced Ca^2+^ signals in several cellular models [109,117,118,119,128]. Second, gene silencing, gene knockdown or dominant negative TPC mutants suppressed NAADP-induced Ca^2+^ signals and downstream signalling events [117,120,128,129,130,131]. Third, direct electrophysiological recordings carried out by using the patch-clamp technique on enlarged lysosomes or by exploiting the planar lipid bilayer technology confirmed that NAADP was able to gate TPCs [124,132,133,134,135,136]. Notably, electrophysiological analysis confirmed that TPC1-2 were both permeable to Ca^2+^ although at a different extent [124,126]. TPC1 is more permeable to monovalent cations, the highest permeability being for H^+^: H^+^ >> K^+^ > Na^+^ ≥ Ca^2+^ [133]. This feature suggests that TPC1 may provide a pathway for intraluminal H^+^ efflux, thereby resulting in alkalization of acidic organelles [18]. Conversely, TPC2 is slightly more selective for Ca^2+^ over monovalent cations and shows a Ca^2+^/K^+^ permeability ratio (PCa^2+^/PK^+^) of 2.6 [132]. It should, however, be pointed out that the ion selectivity and single-channel conductance of TPC1-2 may vary depending on the experimental conditions, although it is nowadays accepted that they mediate Ca^2+^ release [13,124,126].

The discovery that TPCs are Ca^2+^-permeable EL channels targeted by NAADP was subsequently challenged by two studies from the same group, which exploited a lysosomal patch-clamp method to propose that TPCs were rather Na^+^-selective channels with poor Ca^2+^ permeability activated by PI(3,5)P_2_ [137,138]. Several excellent reviews recently addressed this discrepancy that has a remarkable impact on our knowledge of EL Ca^2+^ signaling and associated Ca^2+^-dependent processes. [103,123,125,126,139]. Briefly, by using the same methodology, Jha and coworkers confirmed that TPC2 was permeable to Na^+^ and activated by PI(3,5)P_2_ [135]. Nevertheless, TPC2-mediated currents could also be triggered by NAADP and were sensitive to Mg^2+^: cytosolic Mg^2+^ fully inhibited NAADP-induced currents, while lysosomal Mg^2+^ reduced them by about 20% [135]. Notably, unlike previous reports [137], this study confirmed that TPC2 bears a measurable Ca^2+^ permeability and effectively mediates intracellular Ca^2+^ release in a Mg^2+^-dependent manner [135]. Thus, different cytosolic/lysosomal Mg^2+^ levels could account for the different NAADP-sensitivity displayed by TPC2 in independent studies [123,125]. Accordingly, the studies which neglect the role of TPCs in NAADP-induced Ca^2+^ release routinely used 2 mM Mg^2+^ in their cytosolic solutions [137,138], which is predicted to fully abolish NAADP-mediated lysosomal currents [135]. Moreover, two additional investigations evaluated the Na^+^/Ca^2+^ permeability (PNa^+^/PCa^2+^) ratio of TPCs-mediated currents and found that it was very close to the unity, ranging between 0.86 [129] and 1.1 [133]. Additionally, NAADP does not directly bind to TPCs, but requires the interposition of auxiliary cytosolic proteins [140] that could be diluted or washed away under different in vitro conditions, such as channel reconstitution in planar lipid bilayers or lysosome isolation. To further support the role of TPCs as truly NAADP-gated channels, several concerns were also raised as regard to the validity of the knockout models that were generated to demonstrate that the Ca^2+^ response to NAADP was still present upon genetic deletion of TPCs [123,125,139]. While lack of mRNA or protein expression was not demonstrated in the earlier *Tpcn1^−^*^/−^*/Tpcn2^−/−^* mice [137], Ruas and coworkers revealed that NAADP did not induce Ca^2+^ signals in embryonic fibroblasts deriving from mice demonstrably lacking TPC1, TPC2 or both [129]. Moreover, the Ca^2+^ response to NAADP was rescued by re-expressing wild-type, but not pore-mutant, TPCs [129]. Intriguingly, Ruas and coworkers showed that NH_2_-terminal truncated forms of TPC1 and TPC2 were expressed in the mouse line formerly presented as TPCs-null and were able to rescue the Ca^2+^ response to NAADP in their truly knocked out TPCs model [129,141].

Based on these observations, it is now acknowledged that TPC1-2 serve as NAADP-gated EL Ca^2+^-releasing channels that are co-regulated by PI(3,5)P2 and also permeable to monovalent cations, including H^+^ [103,124]. Moreover, TPCs are able to integrate multiple signals emanated from both the cytosol and the EL luminal although TPC1 and TPC2 show different sensitivity to diverse modulators. For instance, TPC1 (as well as TPC3) was reported to be voltage-activated and to control the electrical excitability of EL vesicles [142], which is in agreement with the presence of two voltage-sensors in TM4 and TM10 in the channel protein. A subsequent study argued that membrane-depolarization is not required for channel activation but increases the affinity of TPC1 for NAADP [134], while TPC2 is voltage-insensitive [132,142]. As mentioned earlier, TPC2 is inhibited by Mg^2+^ [135], whereas there are no information available regarding TPC1 [125]. The regulation by intraluminal pH is more complex. As intuitively expected, some studies revealed that TPC1-2 were both activated by the acidic intraluminal pH [133,134,136]. Nevertheless, TPC1-2 were also shown to be inhibited [132,142] or unaffected by acidic pH [135,137]. Similar to InsP_3_Rs and RyRs, TPC1, but not TPC2 [132], is also regulated by cytosolic Ca^2+^ [133] and could be stimulated by ER-released Ca^2+^ [18]. Luminal EL Ca^2+^, in turn, increases TPC2 sensitivity to NAADP [132], while its stimulatory effect on TPC1 is still matter of controversy [133,134]. Finally, TPC2 activity is increased by p38 and JNK kinases and is finely tuned by EL PI(3,5)P_2_ levels [135]. Likewise, TPCs are stimulated by leucine-rich repeat kinase 2 (LRKK2) [143]. TPC modulation by mTOR will be illustrated in the following paragraph.

##### 2.4.2.1. The Physiological Role of TPCs: Local vs. Global Ca^2+^ Signals

Emerging evidence supports the notion that TPCs may regulate cellular fate and behavior through both local and global intracellular Ca^2+^ signals [13,80,103,123]. TPCs control EL membrane trafficking associated with exocytosis, endocytosis, autophagy and viral infection through an elevation in surrounding Ca^2+^ levels [103,144,145]. For instance, local Ca^2+^ release drives homo- and heterotypic fusion between LEs and lysosomes in fibroblasts isolated from patients affected by Parkinson’s disease (PD) [146]. Lysosomes are enlarged and aggregated in these cells, a morphological defect that can be restored by genetic or pharmacological inhibition of TPC2 and by buffering local Ca^2+^ signals with BAPTA, but not with the slower Ca^2+^ chelator, EGTA [146]. Likewise, TPC2 mediates the fusion events underlying the trafficking to the acidic compartment of several cargoes, including those containing Ebola virus [147], cholera toxin [119], LDL-cholesterol, and EGF/EGF receptor [148]. Notably, local Ca^2+^ release through TPC1 was subsequently shown to promote LEs/lysosomes fusion in normal fibroblasts and HeLa cells [149]. As more widely discussed in [80,103], spatially-restricted Ca^2+^ signals could be generated by TPCs also to mediate retrograde trafficking from endosomes to Golgi [119,150] and secretion of lysosome-related organelles, such as cytolytic granules in cytotoxic T lymphocytes [130]. Two proteomic screens revealed that TPCs may physically interact with a variety of proteins implicated in EL vesicles trafficking and fusion events, including several Rab GTPases, e.g., Rab7A, and syntaxins, e.g., syntaxin 12 and syntaxin 13 [148,151]. Rab7 GTPase in turn boosts NAADP-induced Ca^2+^ release through TPC2 to promote lysosomal enlargement [146,151], whereas syntaxins mediate Ca^2+^-dependent membrane fusion events [152], although it is not known whether syntaxin 12 and syntaxin 13 are *per se* Ca^2+^-sensitive.

In addition, TPCs may regulate autophagy. Unlike TRPML1, mTOR is part of TPC interactome [151] and, under nutrients-rich conditions, it inhibits both the PI(3,5)P_2_-induced TPCs-mediated Na^+^-selective current [138] and NAADP-induced Ca^2+^ release through TPC2 [153]. As a consequence, TPCs are likely to be, partially or fully, inhibited by nutrient supply and, to further support this notion, they are also blocked by ATP [138]. Notably, TPC1-induced Ca^2+^ release is also able to promote the nuclear translocation of the autophagic transcription factor, TFEB [154] (Figure 3). This pathway has recently been shown to protect against lipopolysaccharide-induced liver injury [155]. Several independent studies confirmed that the NAADP/TPCs axis was involved in autophagy, although its role could be more complex than envisaged by earlier work. For instance, NAADP-induced TPC2 activation caused an increase in the number of acidic vesicular organelles and in the levels of the autophagic markers, beclin-1 and LC3II, in rat astrocytes [156], whereas NAADP-evoked Ca^2+^ signals through TPC1 and TPC2 sustained glutamate-induced autophagy in SHSY5Y neuroblastoma cells [157]. Likewise, TPC1 and TPC2 were required for basal and starvation-induced autophagy also in mouse cardiac myocytes [158]. Furthermore, it has been reported that LRKK2 can induce EL Ca^2+^ release through TPCs, thereby recruiting the Ca^2+^/CaM-dependent protein kinase kinase β (CaMKKβ), which initiates the autophagic flux by phosphorylating and activating AMPK [143]. Conversely, TPCs may inhibit autophagy in response to other cellular stressors and/or in other cellular models. For instance, autophagy was exacerbated in starving myoblasts isolated from *TPCN2^−/−^* mice and treated with colchicine to inhibit lysosomal trafficking [159]. It has been proposed that TPC2 negatively modulates autophagy by inducing lysosomal alkalization and reducing protease activity [159].

In addition to local Ca^2+^ signals, NAADP may stimulate TPCs to cause global elevations in [Ca^2+^]_i_ [13,14]. Accordingly, EL Ca^2+^ release can provide the Ca^2+^ trigger necessary to recruit InsP_3_Rs and RyRs, located at the MCSs between EL vesicles and ER cisternae, through the CICR process [35,40,121,122,160,161,162,163]. ER-dependent Ca^2+^ mobilization, in turn, could stimulate NAADP-induced Ca^2+^ release through either local NAADP production or TPC sensitization by cytosolic/intraluminal Ca^2+^ (please, note that only TPC1 is sensitive to intraluminal Ca^2+^) [18]. Ultrastructural analysis revealed that the EL system can be closely (<30 nm) apposed to the ER network [163,164]. As reviewed elsewhere, endosomes-ER MCSs control lipid exchange, cargo sorting, endosome positioning, trafficking and fission [164,165]. Notably, TPC1 favors the formation of these junctional microdomains, thereby promoting the internalization of EGF receptors and curtailing EGF signalling [149,166]. MCSs can be established also between lysosomes and ER to regulate lysosomal positioning [39,167] and mathematical modeling confirmed that these lysosomal-ER microdomains are able to drive the global Ca^2+^ response to the lysosomotropic agent, GPN [39]. Therefore, as NAADP synthesis may occur downstream of GPCRs and TKRs, TPC1-2 may be physiologically engaged to interact with InsP_3_Rs and/or RyRs and control a number of processes [13,51]. These include fertilization [52], neurotransmitter release [168], exocytosis of lytic granules in T cells [130], neurite outgrowth [169], cardiac [35] and vascular smooth muscle [131] contraction, angiogenesis [170], vasculogenesis [171], and nitric oxide release [172,173].

#### 2.4.3. Other endolysosomal Ca^2+^ releasing channels

As mentioned earlier, Ca^2+^ release from the EL store can also be effected by other members of the TRP superfamily, such as TRPA1 and TRPM2, ATP-gated ionotropic P2X4 receptors and voltage-gated Ca^2+^ channels (Figure 1).

TRPM2 is a Ca^2+^-permeable, non-selective cation channel which primarily resides on the PM, is gated by ADP-ribose (ADPr) and hydrogen peroxide (H_2_O_2_), and positively modulated by NAADP and cyclic ADP-ribose [174]. Additionally, TRPM2 may serve as an ADPr-gated lysosomal Ca^2+^ release channels in mouse pancreatic β cells, in which it mediates H_2_O_2_–induced apoptosis [175], and in mouse dendritic cells, in which it controls maturation and migration [176].

TRPA1 belongs to a distinct sub-family of TRP channels and mediates a Ca^2+^-permeable, non-selective cation current that is activated by noxious cold temperatures and mechanical stimuli and by a wealth of irritant compounds in peripheral somatosensory neurons [177]. Herein, TRPA1 is also located within the EL system, thereby mediating local Ca^2+^ sparks that stimulate vesicle exocytosis and calcitonin gene-related peptide release. It has, therefore, been proposed that EL TRPA1 further increases the excitability of dorsal root ganglia [178].

P2X receptors are homo- or heterotrimers which serve as ATP-gated ionotropic, Ca^2+^-permeable, non-selective channels [179]. Seven subunits (P2X1-7) were described in vertebrates, each comprising two TM domains connected by a large extracellular domain, which contains the ATP-binding site, and with cytosolic NH_2_- and COOH-terminal tails [179]. Earlier work revealed that P2X4 receptors were mainly located to the lysosomal vesicles of several cell types, including rat microglia, vascular endothelial cells and freshly isolated peritoneal macrophages, through a dileucine type motif within their NH_2_-terminus [180]. Nevertheless, P2X4 receptors can be recruited to the PM or to the phagosome surface in response to different cellular stimuli [180]. A subsequent report demonstrated that lysosomal P2X4 receptors may be gated by intraluminal ATP in a pH-dependent manner: P2X4 activity was increased by lumen alkalization, while it was inhibited by a reduction in intraluminal pH [181]. Notably, the activation of P2X4 receptors stimulated EL membrane fusion in a CaM-dependent manner [182].

Voltage-gated Ca^2+^ channels mediate extracellular Ca^2+^ entry in response to membrane depolarization to regulate multiple functions, including muscle contraction and gene expression (Ca_V_1), neurotransmitter release (Ca_V_2), and repetitive firing in sino-atrial cells and thalamic neurons (Ca_V_3) [183]. Surprisingly, a recent study reported that Ca_V_2.1, which mediates voltage-gated P/Q-type Ca^2+^ currents at cerebellar terminal synapses and is encoded by the *CACNA1A* gene [183], is also localized to the lysosomes in mouse cultured granule cells [184]. Mutations in Ca_V_2.1 prevented autophagy by inhibiting the fusion between lysosomes and endosomes and/or autophagosomes [184]. Although lysosomal voltage-gated P/Q-type Ca^2+^ currents were not measured, this finding raises the possibility that lysosomal Ca^2+^ release could effectively occur in response to a positive shift in ΔΨ [61].

## 3. The Role of Lysosomal Ca^2+^ Signals in Cancer: Two-Pore Channels Drive Metastasis and Angiogenesis

### 3.1. Lysosomes Contribute to Cancer Hallmarks

The lysosomal membrane serves as a signaling platform that regulates multiple functions, including endocytic traffic, autophagy, and lysosomal function, thereby impacting on the metabolism and fate of the entire cell [33,185]. Therefore, any alteration in lysosomal function or structure may impair vesicular trafficking and ultimately lead to life-threatening diseases. For instance, mutations in many EL genes or defects in the acidic Ca^2+^ content drive or are tightly involved in the development of neurodegenerative disorders, such as MLIV, Parkinson’s disease, Alzheimer’s disease, and Niemann Pick disease, type C [20,33,38,146,186]. In addition, it is now recognized that lysosomes sustain several cancer hallmarks, including aberrant proliferation, metastasis and angiogenesis [185,187,188]. Briefly, lysosomes sustain excessive proliferation by supplying cancer cells with the energy and metabolic precursors necessary for rapid division and mass accumulation, thereby overcoming the limited delivery of nutrients and oxygen to the harsh microenvironment of growing tumors [188,189]. In neoplastic cells, lysosomes heavily recycle exogenous macromolecules and endogenous proteins/organelles to fuel cell growth and, not surprisingly, autophagy plays a crucial role in malignant transformation and cancer progression [188]. Notably, malignant transformation repositions lysosomes from the juxtanuclear pool towards the cell periphery, increases lysosomal volume and ultimately leads to an increase in total protease activity [185,190]. As a consequence, the cathepsins secreted via exocytosis of peripheral lysosomes contribute to pave the way for cancer cell migration and tumor angiogenesis. Accordingly, lysosomal-derived cathepsins may degrade the extracellular matrix components either directly or indirectly by stimulating matrix metalloproteins, which supports primary cancer migration, invasion and metastatic dissemination [185,188]. In parallel, such remodeling of the extracellular matrix is also essential for capillary sprouting and the initiation of angiogenesis, which is indispensable to nourish cancer cells and to let them colonize distant organs [191,192]. Cathepsins D, B, S, K, and L are the main hydrolases supporting the local invasion of primary tumor cells into surrounding tissues and their subsequent intravasation into the circulatory system [185,188]. It has been envisioned that lysosomal exocytosis results in the acidification of local cancer microenvironment, which is necessary for cathepsins to effectively degrade the extracellular matrix [185]. This hypothesis has been supported by the observation that several v-ATPase subunits are up-regulated and expressed on the PM of cancer cells, where they participate in the creation of the acidic extracellular environment which supports a more invasive phenotype [193]. Finally, lysosomes may mediate drug resistance to several chemotherapeutics by exploiting multiple mechanisms, such as passive ion-trapping based sequestration and clearance into the extracellular milieu via exocytosis [194]. It should, however, be pointed out that lysosomes could play an oncosuppressive role during the early stages of the tumorigenic process. We refer the reader to some excellent reviews covering this intriguing issue [190,195].

### 3.2. The Role of Endolysosomal Ca^2+^ Signaling in Cancer: Are Two-Pore Channels the Guilty Ones?

As aforementioned, mutations in EL Ca^2+^-permeable channels, i.e., TRPML1, or defects in the acidic Ca^2+^ store underlie severe neurodegenerative disorders, such as MLIV and Niemann Pick disease, type C, respectively [38,186]. Moreover, rewiring of the intracellular Ca^2+^ toolkit supports many cancer hallmarks. For instance, the reduction in ER Ca^2+^ concentration is responsible for the increased resistance to pro-apoptotic stimuli [196,197] and anti-angiogenic treatments [198,199]. Likewise, the up-regulation of STIM and/or Orai1 proteins, as well as of multiple TRP isoforms, supports uncontrolled proliferation [200,201], tissue invasion and migration [201,202,203], and sustained angiogenesis [204,205]. Therefore, recent studies assessed whether also the EL Ca^2+^ machinery was implicated in malignant transformation [19,20,21]. No evidence has been presented about the involvement of EL TRPML1, TRPM2, P2X4 receptors and voltage-gated Ca^2+^ channels in malignant transformation [19,21]. A recent report showed that lysosomal biogenesis and function were increased in pancreatic ductal adenocarcinoma (PAD) and associated to constitutive nuclear localization of TFEB [206]. Surprisingly, TFEB activity was independent on metabolic pathways in PAD and sustained autophagy-lysosome activation even when mTORC1 was not inhibited [206]. This observation suggests that TFEB could be functionally decoupled by TRPML1 in cancer cells. Enhanced TFEB expression and/or activity also induced autophagy, migration and metastasis in non-small lung cell carcinoma [207] and renal cellular carcinoma [208]. These studies, however, did not assess the Ca^2+^-dependence of the subcellular localization of TFEB in cancer cells. TRPML2 is the only member of this sub-family to have been clearly associated to malignant transformation. A recent investigation revealed that TRPML2 transcript and protein were highly expressed in glioma tissues and that their expression increased during tumor progression [209]. Gene silencing experiments conducted in glioma cell lines showed that TRPML2 regulates proliferation through Akt and ERL 1/2 phosphorylation and prevents caspase-3 activation, thereby protecting glioma cells from apoptosis [209]. The role of Ca^2+^ signalling in the oncogenic effect of TRPML2 was not explored. Conversely, emerging evidence clearly hints at TPCs as main drivers of multiple cancer hallmarks in a growing number of solid malignancies (Figure 3). These evidences are described in the next paragraph. 

### 3.3. Two-Pore Channels 1 and 2 Support Cancer Cell Migration and Promote Tumor Vascularization

Earlier work revealed that TPC1 transcripts were ~3–8 fold higher than TPC2 in SKBR3 cells, a human breast cancer cell line, and in PC12 cells, a model for rat pheochromocytoma [128]. Subsequently, it was found that TPC1 and TPC2 transcripts were similarly expressed in a different set of human breast cancer cells, although only genetic silencing of TPC2 inhibited EGF-induced vimentin, but not E-cadherin, expression in the highly aggressive MDA-MB-468 cell line. On the other hand, silencing of either TPC isoform did not affect the rate of cell proliferation [210]. TPC2 silencing reduced the amount of Ca^2+^ released upon pharmacological depletion of the ER under 0 Ca^2+^ conditions and the amplitude of SOCE recorded upon restoration of extracellular Ca^2+^ levels [210]. Notably, TPC2 may interact with STIM1 and Orai1 in response to ER Ca^2+^ depletion in the megakaryoblastic cell line MEG01 [211]. More recently, TPC1 and TPC2 transcripts were identified in several cancer cell lines established from liver (HUH7), bladder (T24) and blood (Jurkat) as compared to a human breast cancer cell line (MDA-MB-231) [212]. Overall, TPC1 transcripts were more abundant than TPC2 in T24 and HUH7 cells, whereas immunocytochemistry revealed that TPC2 protein was up-regulated in liver carcinoma as compared to adjacent healthy tissue [212]. Electrophysiological recordings confirmed that EL currents were activated by PI(3,5)P_2_ and inhibited by tetrandrine, a rather selective TPC inhibitor [212]. Genetic silencing and pharmacological blockade of TPC1 and TPC2 with tetrandrine or NED-19, another widely employed TPC inhibitor, blocked adhesion and migration in invasive cancer cells (T24 and HUH7) [205]. Furthermore, genetic silencing of TPC2 and pharmacological blockade of TPCs with tetrandrine or NED-19 prevented lung metastasis in a mouse model of mammary carcinoma [212]. The same investigation disclosed that TPCs inhibition caused β1-integrin accumulation within EEs and impaired its trafficking towards the PM, thereby halting lamellipodia formation and interfering with the migration process [212]. Migrating cells extend lamellipodia in the direction of migration, which need to be stabilized through integrins bound to the actin cytoskeleton [205]. Genetic and pharmacological inhibition of TPCs prevented the formation of β1-integrin-, pFAK-, pSrc-, and vinculin-positive polarized lamellipodia [212]. As pointed out in [20], deletion of the recently identified nonplacental mammalian CAX in neural crest cells impaired spreading and focal adhesion formation in vitro, thereby impairing migration in vivo [54]. Notably, this study suggested that local, rather than global, cytosolic Ca^2+^ signals were required to sustain motility as the same effect was phenocopied by BAPTA, while EGTA was ineffective [54]. Therefore, EL Ca^2+^ signals could effectively modulate migration in both normal and neoplastic cells. Conversely, TPC2 negatively modulates EL trafficking in 4T1 mouse breast cancer cells and HeLa mouse cervical cancer cells by preventing the recruitment of Rab7 from lysosomes to autophagosomes and causing an increase in lysosomal pH [213]. As mentioned earlier, autophagy tends to suppress the early stages of many solid tumors. One could envisage that, under these circumstances, TPC overexpression could favor tumor progression.

The first report about the pro-tumorigenic role of TPCs was, however, provided by Favia and coworkers [170]. This study revealed that NAADP-induced Ca^2+^ signals promoted tumor development in mice xenografted with B16 melanoma cells [214]. Accordingly, the pharmacological blockade of TPCs with NED-19 halted melanoma growth and vascularization and significantly reduced lung metastasis [214]. A more insightful analysis disclosed that NAADP stimulated melanoma B16 cells to migrate and proliferate through an increase in [Ca^2+^]_i_ that was sustained by the EL Ca^2+^ store [170]. NAADP-evoked Ca^2+^ signals induced Go/G1 progression and the activation of the focal adhesion kinase (FAK), which is indispensable for cell migration [170].

This study further suggested that the Ca^2+^ response to NAADP stimulated angiogenesis through the paracrine release of growth factors from the cancer cells. In addition, the same group previously reported that NAADP sustains vascular endothelial growth factor (VEGF)-dependent angiogenesis in vitro and neovessel formation in vivo [170]. Accordingly, they found that VEGF-induced intracellular Ca^2+^ signals were attenuated by NED-19 and v-ATPase inhibition with bafilomycin A1. Likewise, the pharmacological blockade of TPCs with NED-19 blocked cell proliferation, migration and tube formation by suppressing VEGF-induced phosphorylation of ERK1/2, Akt, JNK and endothelial nitric oxide synthase (eNOS) [170] (Figure 4). Finally, the Matrigel plug assay revealed that VEGF-induced angiogenesis in vivo was blocked by NED-19 in wild type mice and in *TPCN2^−/−^*, but not *TPCN1^−/−^*, mice [170]. These results were further supported by the finding that also the natural flavonoid, naringenin, attenuated VEGF-induced angiogenesis in vitro and neovessel formation in vivo [215]. Accordingly, naringenin blocked TPC1- and TPC2-mediated lysosomal currents, significantly reduced VEGF-induced Ca^2+^ signals in HUVECs and prevented their assembly into capillary-like bidimensional structures in Matrigel scaffolds [215]. Furthermore, naringenin halted VEGF-induced vascularization of Matrigel plugs subcutaneously implanted in mice [215]. In addition, a functional EL Ca^2+^ store enriched with TPC1 is also expressed in human endothelial colony forming cell (ECFCs) [41,171], a subset of endothelial progenitor cells that support tumor vascularization by physically engrafting within neovessels and through paracrine release of pro-angiogenic growth factors [198,216]. Notably, liposomal delivery of NAADP promoted ECFC proliferation by likely recruiting eNOS in a Ca^2+^/CaM-dependent manner [41,171]. Tumor-derived ECFCs are largely insensitive to VEGF [198,204,217,218,219]. This feature raises the intriguing possibility that TPC1 could provide an alternative target to VEGF signalling to prevent or, at least attenuate, ECFC homing to tumor neovessels.

### 3.4. TPCs as Therapeutic Target in Cancer

The studies described strongly suggest that TPC1 and TPC2 represent suitable molecular targets to interfere with tumor growth, vascularization, and metastasis. Pharmacological blockade of TPCs could thus be regarded as a promising, alternative anticancer approach [19,21]. Throughout the present article, we have already introduced a number of compounds that are routinely employed to block TPCs and their associated functions. These include NED-19, which is likely to prevent NAADP binding [220], naringenin, tetrandrin and structurally-related derivatives, that are likely to plug the channel pore [147,221]. Moreover, the nucleotide mimetics, PPADS and PPNDS, may also be exploited as competitive antagonists of NAADP receptors [222], i.e., TPC1 and TPC2. Intriguingly, it has long been known that the Ca^2+^ response to NAADP was inhibited by antagonists of L-type Ca_V_ channels, such as verapamil and diltiazem and dihydropyridines (e.g., nifedipine, nisoldipine, nimodipine and nicardipine) [223,224]. Subsequent work revealed that TPCs present a binding site for Ca_V_ antagonists within their selectivity filter, which is not surprising when considering the strong molecular similarity between TPCs and Ca_V_ [110]. As reviewed elsewhere [225], Ca_V_ antagonists, such as nitrendipine, nifedipine and verapamil, are routinely employed to treat life-threatening disorders [9], such as arrhythmia, infarction-induced cardiac failure, hypertension and chronic stable angina, and lack severe off-target effects. This observation suggests that Ca_V_ antagonists could also be probed in preclinical trials assessing whether they are able to induce cancer shrinkage by blocking TPCs.

A larger arsenal of drugs is currently under intense scrutiny to design alternative treatments of diseases associated to mutated or dysfunctional TPCs, including neurodegenerative diseases, or requiring a functional EL trafficking, such as viral infections. A recent screening campaign conducted on sea urchin egg homogenates disclosed a novel pharmacopeia of 18 drugs effectively (>80%) inhibiting NAADP-induced Ca^2+^ release. The most efficient inhibitors were PF-543, a cell permeant blocker of sphingosine kinase 1, SKF96365, a non-specific antagonist of Ca^2+^-permeable channels that was already known to affect NAADP signalling [226], and racecadotril, a neutral endopeptidase inhibitor [227]. The subsequent validation on a human cell line, i.e., U2OS, confirmed that these drugs were able to inhibit NAADP-induced Ca^2+^ signals [227] and attenuated the translocation of a Middle East Respiratory Syndrome Coronavirus pseudovirus through the EL system [221]. Parallel work screened a database of ≈1500 Food and Drug Administration (FDA)-approved drugs in search for therapeutically relevant TPC2 inhibitors [228]. This throughout analysis revealed that four dopamine receptor antagonists and five selective estrogen receptor modulators were able to bind to TPC pore. Accordingly, these drugs reduced the mean open probability of TPC2 rerouted to the PM and inhibited lysosomal TPC2 currents [228]. Notably, these drugs inhibited infection of HeLa by Ebola virus [228]. Finally, a battery of small-molecule NAADP analogues were recently prepared by alkylation of nicotinic acid derivatives with a series of bromoacetamides and tested on T cells- and cardiomyocytes-related disorders [229]. One of these agents, termed lead compound 2, inhibited NAADP-induced Ca^2+^ release in rat Jurkat T-lymphocytes in vitro and was effective in an EAE model of multiple sclerosis in vivo. Another agent obtained by this screening, termed 3 or BZ194 [229], blocked the Ca^2+^ response to NAADP in vitro [230] and cardiac arrhythmias evoked by β-adrenergic stimulation in vivo [231].

Only NED-19, tetrandrin and naringenin were hitherto probed to halt tumor growth and vascularization. Unfortunately, tetrandrine and naringenin may target multiple signalling pathways and are not approved for human use worldwide [228]. However, the effort to identify novel drugs to effectively treat a number of diseases associated to EL Ca^2+^-permeable channels, such as TPCs, is now coming of age and is predicted to rapidly enlarge our arsenal of selective TPC inhibitors, which could be also testable for anticancer treatments.

## 4. Conclusions

EL Ca^2+^ signalling through TRPML1-3 and TPC1-2 finely regulates cellular fate and behavior by driving the EL trafficking machinery and the autophagic flux, thus enabling the cells to adapt to stressful conditions, such as those imposed by lack of nutrients. In addition, local EL Ca^2+^ signals control trafficking and correct positioning of signaling proteins involved in cell proliferation and migration, such as integrins and EGF receptors, and the endocytosis of bacteria, toxins and viruses, which may cause cell death. Moreover, TPCs may interact with ER-embedded InsP_3_Rs and RyRs to generate global Ca^2+^ signals that further expand the repertoire of cellular functions regulated by the EL Ca^2+^ store. Herein, we discussed the evidences supporting the role of EL Ca^2+^ signalling in cancer. While the oncogenic function of TRPML1, the main driver of TFEB activation in healthy cells, remains obscure, TPC1 and TPC2 are standing out as crucial drivers of many cancer hallmarks, including excessive proliferation, aberrant migration and sustained vascularization. Many issues remain to be elucidated. For instance, is TPC expression actually altered in cancer cells? Is NAADP the main ligand of tumor TPCs or is there a role also for PI(3,5)P_2_? Is the EL Ca^2+^ store altered by neoplastic transformation, just as it happens for the ER? Which is the main source of Ca^2+^ responsible for EL Ca^2+^ refilling in cancer cells: the cytosol, the ER or the extracellular milieu? Does CICR support the Ca^2+^ response to NAADP also in cancer cells (Figure 3)? With all these questions awaiting an answer, the field is surely going to open its doors to much exciting research.

## Figures and Tables

**Figure 1 cancers-11-00027-f001:**
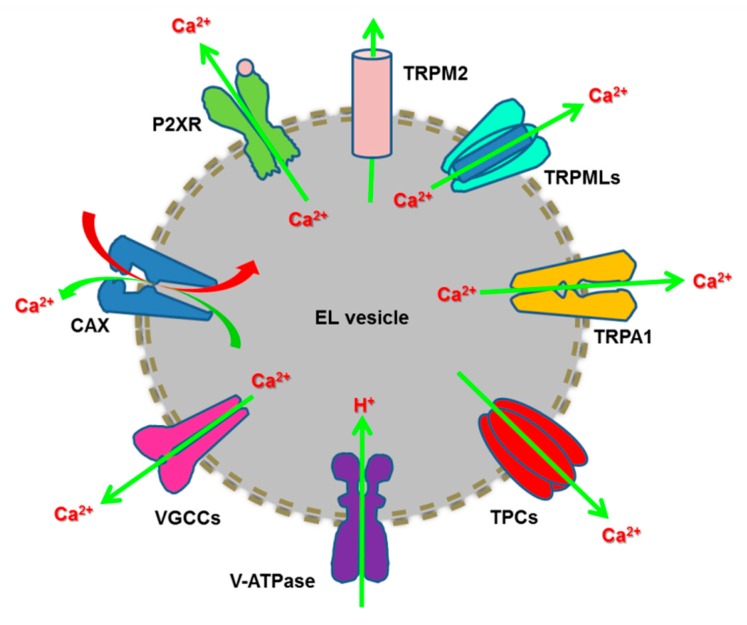
The endolysosomal Ca^2+^ toolkit. Refilling of the EL Ca^2+^ store is driven by the H+ gradient established through the v-ATPase and effected by a Ca^2+^/H^+^ exchanger (CAX) in nonplacental mammalian cells. EL Ca^2+^ release may occur through multiple Ca^2+^-permeable channels, such as such as TRPML1-3, TRPM2, TRP Ankyrin 1 (TRPA1), TPC1-2, ATP-gated ionotropic P2X4 receptors and voltage-gated Ca^2+^ channels (VGCCs).

**Figure 2 cancers-11-00027-f002:**
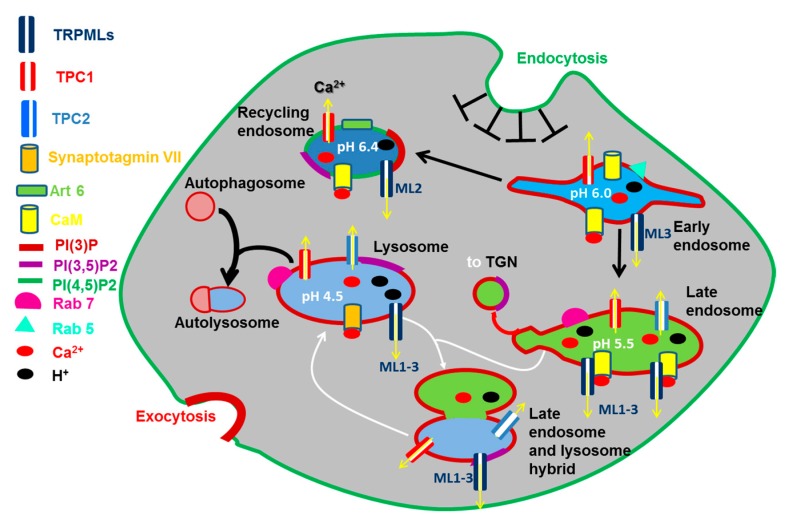
TRPML1-3 and TPC1-2 in the endocytic pathway. Endosomal vesicles undergo cargo-dependent maturation (black arrows), membrane fusion (white arrows), and fission/budding (red arrow). The molecular identity of each intracellular compartment is defined by specific recruitment of small G proteins (Rab and Arf GTPases) and the composition of phosphoinositides (PIPs). Endolysosomal compartments are featured by a weak-to-highly acidic pH, which has therefore been indicated for each organelle. Many steps of the endocytic pathway are Ca^2+^-dependent due to the involvement of specific Ca^2+^-dependent decoders, such as calmodulin (CaM) and synaptotagmin VII (Syt), and Arf6. All the abbreviations are indicated on the left, with the exception of TGN (Trans Golgi Network).

**Figure 3 cancers-11-00027-f003:**
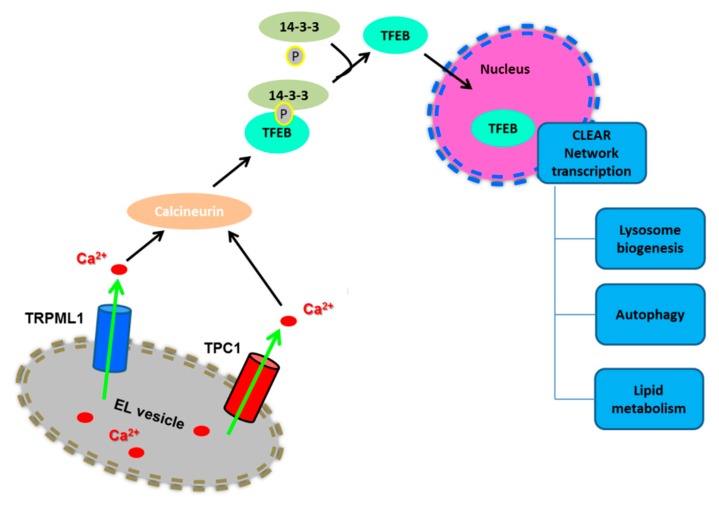
Lysosomal Ca^2+^ signaling induces the nuclear translocation of TFEB. Lysosomal Ca^2+^ release through TRPML1 or TPC1 stimulates the nuclear translocation of TFEB to activate the CLEAR network transcription. Please, see the text for further details.

**Figure 4 cancers-11-00027-f004:**
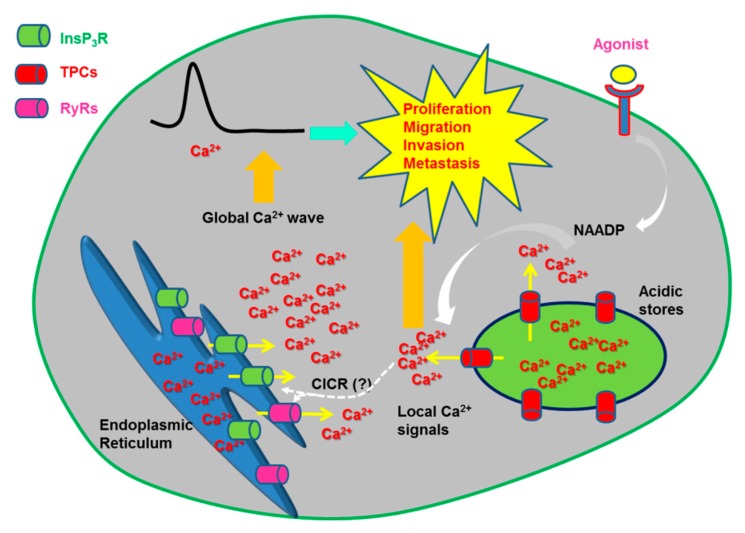
TPCs mediate oncogenic Ca^2+^ signals. Emerging evidence indicates that NAADP promotes tumorigenesis by inducing EL Ca^2+^ release through TPCs. Work carried out on melanoma B16 cells suggest that NAADP is synthesized in response to VEGF stimulation. NAADP-induced Ca^2+^ signals promote tumorigenesis by stimulating migration, invasion, angiogenesis and, maybe, proliferation. The evidence that NAADP-induced local Ca^2+^ release through TPCs is amplified in a global elevation in [Ca^2+^]_i_ through the Ca^2+^-induced Ca^2+^ release (CICR) process is yet to be provided.

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
