# Peer review of "Endolysosomal Ca2+ Signalling and Cancer Hallmarks: Two-Pore Channels on the Move, TRPML1 Lags Behind!"

_cancers, 2018, doi:10.3390/cancers11010027_

Reviewer 1 Report

This is a well written and thorough review on the mechanisms of acidic Ca2+ store signaling and the resulting pathological significance for cancer progression. The authors have done a very good job at summarizing the field and relevant recent literature. I feel this should be a well cited review article by the field, and I very much enjoyed reading this article.

I have only minor suggestions – first – Figures 2 and 3 in the .pdf are too small to read the associated figure labelling; second there is a typo on line 181 (protonmotRive force).

Author Response

Dear Reviewer #1,

We are gratefully thankful for your comments on our manuscript entitled: “Endolysosomal Ca2+ signalling and cancer hallmarks: Two-pore channels on the move, TRPML1 lags behind!” for publication as Review Article in Cancers – Special Issue Ion channels in cancer.

We enlarged the labels in Figure 2 and in Figure 3 (now Figure 4) and corrected the typo you highlighted on line 181 (now proton-motive force).

We thank you again for the nice comments on our manuscript.

Sincerely,

Francesco Moccia

Francesco Moccia, PhD

Laboratory of General Physiology,

Department of Biology and Biotechnology “L. Spallanzani”

University of Pavia,

Via Forlanini 6, 27100, Pavia, Italy.

Tel: 0039 0382 987614.

Fax: 0039 0382 987527.

Reviewer 2 Report

 The authors describe the emerging role od endolysosomal Ca2+ channels, belonging to the Transient Receptor Potential and Two Pore channels in cancers.

Major points: the paper is well written, the English is correct and the References are on line. However we think that the section reporting basic informations relative to endolysosomal channel functions, from pg1 to pg 10, are too long as compared to the part relative to the goal of this review. In addition, I think that the authors must to add 1 or 2 new figures, more explicative relative to this section, for better use of the readers.

Minor points:

Pg. 4 line 141: the concept relative to lysosome as "hub" of mulple integrated signals, is very interesting. This information requires more details. New data and references must to be added.

Pg. 5 line 160: TFEB has been found to activates CLEAR gene network. This information require more informations and references.

Pg. 13 line 602: the contribute of NAADP to FAK activation in cell migration need to be further explained.

Some abbreviation of References are lacking

Author Response

Dear Reviewer #2,

We are gratefully thankful for your comments on our manuscript entitled: “Endolysosomal Ca2+ signalling and cancer hallmarks: Two-pore channels on the move, TRPML1 lags behind!” for publication as Review Article in Cancers – Special Issue Ion channels in cancer.

We amended the manuscript according to your suggestions and strongly believe that your comments remarkably improve the quality of our work.

More specifically:

Major points: the paper is well written, the English is correct and the References are on line. However we think that the section reporting basic informations relative to endolysosomal channel functions, from pg1 to pg 10, are too long as compared to the part relative to the goal of this review. In addition, I think that the authors must to add 1 or 2 new figures, more explicative relative to this section, for better use of the readers.

We thank the referee for this comment. We added an additional figure, i.e. Figure 3, to illustrate how TRPML1 and TPCs stimulate the nuclear translocation of TFEB. We also understand the Referee’s comment on manuscript length. However, we received very positive comments on the wealth of information reported by the other two Reviewers, one of whom even suggested us to provide additional information. Therefore, we thought it was reasonable to leave the paragraphs related to endolysosomal Ca2+ release and refilling as they are.

Pg. 4 line 141: the concept relative to lysosome as "hub" of mulple integrated signals, is very interesting. This information requires more details. New data and references must to be added.

We thank the Referee for this very nice comment. We expanded upon this sentence and add more references (Page 4, Line 147).

Pg. 5 line 160: TFEB has been found to activates CLEAR gene network. This information require more informations and references.

This is absolutely correct and we provided this information on Page 5, Line 310.

Pg. 13 line 602: the contribute of NAADP to FAK activation in cell migration need to be further explained.

This observation is also absolutely sound and we expanded upon NAADP-induced FAK activation on Page 14, Line 773.

We do hope that you will now regard our manuscript suitable for publication on Cancers.

Sincerely,

Francesco Moccia

Francesco Moccia, PhD

Laboratory of General Physiology,

Department of Biology and Biotechnology “L. Spallanzani”

University of Pavia,

Via Forlanini 6, 27100, Pavia, Italy.

Tel: 0039 0382 987614.

Fax: 0039 0382 987527.

Reviewer 3 Report

The manuscript was well written and was easy to follow. It covered all the new development in the field. However, I would suggest the authors consider the following comments before publishing the paper.

Specific comments:

Figure 1: I guess TRPML1 should be TRPMLs; TRPM2 is missing.

Figure 2: It is hard to differentiate some of the molecules with similar colors such as TRPMLs-TPC1 and PI3P-PI3,5P2.

Page 4: The rationale for mTOR is not clear. Something to do with lysosomal Ca2+? In addition, mTOR part should be in a separate paragraph.

Page 5: BK channel in endolysosmal Ca2+ refilling need to be included.

Page 6, line 239: TRPML1 is not H+ permeable. This need to be corrected.

Page 6, line 253: a new paper regarding TRPML1-mTOR (Autophagy. 2018;14(1):38-52) need to be included.

Page 6-7: TRPML1 may control autophagy using three mechanism: TFEB, autophagosome-lysosome fusion (Nat Cell Biol. 2016 Apr;18(4):404-17), and mTOR regulation (Autophagy. 2018;14(1):38-52). All these need to be discussed.

P7, line 282: reference for PI4,5P2 inhibiting TRPML3.

P7, line 289:  TRPML1 can also direct TRPML3 targeting to endolysosomes.

Author Response

Dear Reviewer #3,

We are gratefully thankful for your comments on our manuscript entitled: “Endolysosomal Ca2+ signalling and cancer hallmarks: Two-pore channels on the move, TRPML1 lags behind!” for publication as Review Article in Cancers – Special Issue Ion channels in cancer.

We amended the manuscript according to your suggestions and strongly believe that your comments remarkably improve the quality of our work.

More specifically:

Figure 1: I guess TRPML1 should be TRPMLs; TRPM2 is missing.

You were quite right and the figure is modified according to your observations.

Figure 2: It is hard to differentiate some of the molecules with similar colors such as TRPMLs-TPC1 and PI3P-PI3,5P2.

We thank the Referee for this comment and we changed the colors as suggested.

Page 4: The rationale for mTOR is not clear. Something to do with lysosomal Ca2+? In addition, mTOR part should be in a separate paragraph.

This observation is quite good and we reworded the text accordingly. In particular, we add a new paragraph to explain the role of lysosomal Ca2+ in mTOR activation. As the Referee realized, the rationale for the previous organization of the text was the Ca2+-dependency of mTOR, but this concept was complicated to be inferred. We thank the Referee for this suggestion.

Page 5: BK channel in endolysosmal Ca2+ refilling need to be included.

We followed your suggestion and described the role of BK channels in lysosomal Ca2+ signalling on Page 6, Line 368.

Page 6, line 239: TRPML1 is not H+ permeable. This need to be corrected.

We thank the referee for this observation! We changed the text accordingly (Page 7, Line 393).

Page 6, line 253: a new paper regarding TRPML1-mTOR (Autophagy. 2018;14(1):38-52) need to be included.

We thank the referee for this suggestion and included the reference in the text (Page 8, Line 438).

Page 6-7: TRPML1 may control autophagy using three mechanism: TFEB, autophagosome-lysosome fusion (Nat Cell Biol. 2016 Apr;18(4):404-17), and mTOR regulation (Autophagy. 2018;14(1):38-52). All these need to be discussed.

We thank the referee for this suggestions and described all the three mechanisms on Page 8.

P7, line 282: reference for PI4,5P2 inhibiting TRPML3.

Actually, we think this was a typo due to some copy and paste operation. As far as we know, TRPML3 is not reportedly inhibited by PI(4,5)P2. We do thank you for raising this issue.

P7, line 289:  TRPML1 can also direct TRPML3 targeting to endolysosomes.

We thank you for this suggestion and discussed this issue on Page 6, Line 384.

We do hope that you will now regard our manuscript suitable for publication on Cancers.

Sincerely,

Francesco Moccia

Francesco Moccia, PhD

Laboratory of General Physiology,

Department of Biology and Biotechnology “L. Spallanzani”

University of Pavia,

Via Forlanini 6, 27100, Pavia, Italy.

Tel: 0039 0382 987614.

Fax: 0039 0382 987527.
